# UD-Mamba: A pixel-level uncertainty-driven Mamba model for medical image segmentation

## Abstract

Recent advancements have highlighted the Mamba framework, a state-space models (SSMs) known for its efficiency in capturing long-range dependencies with linear computational complexity. While Mamba has shown competitive performance in medical image segmentation, it encounters difficulties in modeling local features due to the sporadic nature of traditional location-based scanning methods and the complex, ambiguous boundaries often present in medical images. To overcome these challenges, we propose Uncertainty-Driven Mamba (UD-Mamba), which redefines the pixel-order scanning process by incorporating channel uncertainty into the scanning mechanism. UD-Mamba introduces two key scanning techniques: sequential scanning, which prioritizes regions with high uncertainty by scanning in a row-by-row fashion, and skip scanning, which processes columns vertically, moving from high-to-low or low-to-high uncertainty at fixed intervals. Sequential scanning efficiently clusters high-uncertainty regions, such as boundaries and foreground objects, to improve segmentation precision, while skip scanning enhances the interaction between background and foreground regions, allowing for timely integration of background information to support more accurate foreground inference. Recognizing the advantages of scanning from certain to uncertain areas, we introduce four learnable parameters to balance the importance of features extracted from different scanning methods. Additionally, a cosine consistency loss is employed to mitigate the drawbacks of transitioning between uncertain and certain regions during the scanning process. Our method demonstrates robust segmentation performance, validated across three distinct medical imaging datasets involving pathology, dermatological lesions, and cardiac tasks.

## 1 Introduction

Transformers have shown significant potential in image processing due to their ability to model long-range dependencies (Vaswani et al., 2017; Dosovitskiy et al., 2021; Liu et al., 2021; Bao et al., 2024; Zhang et al., 2024b). However, their quadratic computational complexity with respect to sequence length imposes substantial computational costs, particularly in high-resolution tasks like medical image segmentation. Recently, state-space models (SSMs) have emerged as a more computationally efficient alternative, offering linear complexity while preserving the ability to model long-range dependencies (Gu et al., 2021). Among these, the Mamba architecture (Gu & Dao, 2023; Dao & Gu, 2024) stands out, employing selective scanning techniques and hardware-optimized design to achieve impressive results across various visual tasks (Liu et al., 2024b; Zhu et al., 2024; Li et al., 2024; Hu et al., 2024; Liu et al., 2024a).

In medical image segmentation, the primary objective is to accurately delineate regions that correspond to target organs or pathological tissues, providing essential support for clinical diagnoses (Ronneberger et al., 2015; Chen et al., 2024; Isensee et al., 2021; Hatamizadeh et al., 2022; Li et al., 2018; Wang et al., 2021). Due to its capacity to capture long-range dependencies and process high-resolution images efficiently, the Mamba framework has seen increasing application in the medical imaging field (Yang et al., 2024; Xing et al., 2024). However, Mamba's traditional position-based sequential scanning method often leads to intermittent scanning of different semantic regions (Figure 1(e)), which is particularly problematic when dealing with complex backgrounds and ambiguous boundaries in medical images. This hinders Mamba's ability to accurately model local features essential for effective segmentation (Fan et al., 2024; Wang et al., 2024a).

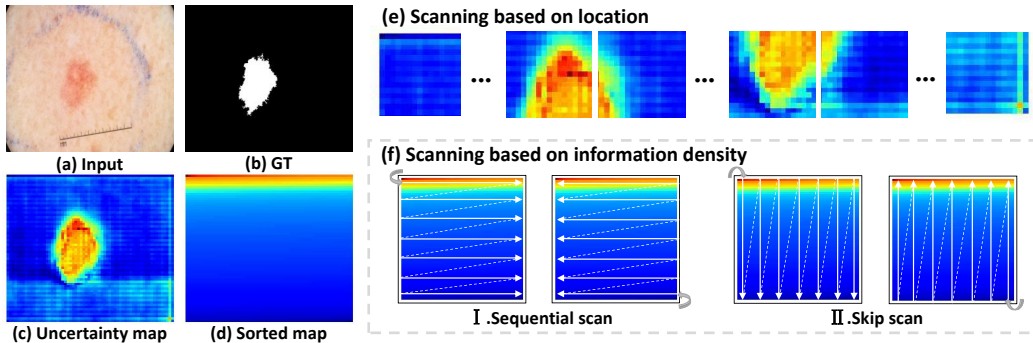

Figure 1: Pixel-level channel uncertainty-based scanning mechanism. (a) Input image; (b) Ground truth; (c) Resulting image obtained from channel-based uncertainty calculations; (d) Feature image sorted by the degree of uncertainty; (e) Previous method using the SS2D scanning mechanism; (f) Our UD-BSS scanning mechanism, which includes sequential scanning and skip scanning.

To overcome this limitation, we propose Uncertainty-Driven Mamba (UD-Mamba), which leverages channel uncertainty as a guiding metric to redefine the pixel-wise scanning process. As illustrated in Figure 1(c), pixels with higher median channel uncertainty are often associated with critical areas, such as the foreground and boundaries. Conversely, regions with lower uncertainty are typically related to the background. By calculating the uncertainty map and ranking the pixels based on their uncertainty levels, as shown in Figure 1(d), we ensure that uncertain (and thus critical) regions are distinguished from more certain regions (typically representing background information).

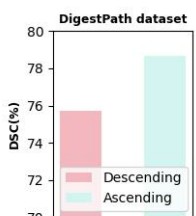

Figure 2: Ascending vs. descending uncertainty.

The proposed scanning strategy, as depicted in Figure 1(f), introduces two key methods: 1) **Sequential scanning**: This method processes pixels in strict order according to their uncertainty levels, effectively clustering high-uncertainty regions such as boundaries and foreground areas. By focusing on these critical regions, sequential scanning ensures that the model captures the fine details in areas crucial for accurate segmentation. 2) **Skip scanning**: This technique moves vertically across the image at consistent uncertainty intervals, enhancing the interaction between background and foreground information. It supplements the model's understanding of background regions while ensuring precise foreground segmentation. By combining sequential and skip scanning, UD-Mamba is able to focus on the fine structures of critical regions while maintaining an understanding of the broader context. This dual-scanning approach enables a more balanced and effective segmentation performance. Furthermore, as illustrated in Figure 2, scanning from regions of low uncertainty to high uncertainty generally yields superior results compared to the reverse order. To optimize this process, we introduce four learnable parameters that adjust the importance of features gathered from different scanning techniques. Additionally, we apply a cosine consistency loss to ensure that features derived from scanning uncertain-to-certain regions are aligned with those from certain-to-uncertain regions, further enhancing segmentation accuracy.

Our contributions can be summarized as follows:

- We introduce a novel pixel-level selective scanning approach guided by channel uncertainty, addressing the limitations of traditional position-based sequential scanning methods.

- We incorporate learnable parameters to balance feature importance across different scanning directions and employ a cosine consistency loss to align forward and backward scan results, improving feature consistency.

- Extensive experiments on three medical imaging datasets demonstrate that UD-Mamba effectively identifies ambiguous regions, leading to more reliable segmentation outcomes compared to existing Mamba-based methods.

## 2 RELATED WORK

### 2.1 MEDICAL IMAGE SEGMENTATION

In medical image segmentation, Convolutional Neural Networks (CNNs) and Transformers dominate as leading frameworks. A significant advancement in CNN-based segmentation was introduced by UNet (Ronneberger et al., 2015), which employs a symmetric encoder-decoder architecture with skip connections. These skip connections effectively integrate local features from the encoder with semantic information from the decoder, setting the foundation for many subsequent improvements (Zhou et al., 2019; Oktay et al., 2018; Le & Saut, 2023). Despite its success, CNN-based methods are limited by their local receptive fields, which hinder the capture of long-range dependencies essential for dense prediction tasks.

Inspired by the Vision Transformers (ViTs) (Dosovitskiy et al., 2021; Liu et al., 2021), there has been increasing interest in incorporating Transformers into medical image segmentation. TransUNet (Chen et al., 2024), one of the pioneering works, introduced a hybrid model that uses Transformers in the encoder to model global context, while retaining the overall UNet structure. Swin-UNet (Cao et al., 2022) further explored a fully Transformer-based framework for segmentation tasks. While Transformers are adept at modeling long-range dependencies, their self-attention mechanism introduces quadratic complexity relative to input size, which poses scalability challenges, especially in pixel-level tasks like medical image segmentation.

### 2.2 MAMBA-BASED MEDICAL IMAGE SEGMENTATION

State Space Models (SSMs) have recently emerged as a powerful tool for visual tasks, with Mamba (Gu & Dao, 2023; Dao & Gu, 2024) showing promising results by efficiently modeling global context with linear complexity. Mamba-based models have demonstrated their versatility across a range of applications (Zhu et al., 2024; Ruan & Xiang, 2024; He et al., 2024; Zhang et al., 2024a; Fan et al., 2024). U-Mamba (Ma et al., 2024) introduces a hybrid framework combining CNNs and SSMs, effectively capturing both local and global features. Swin-UMamba (Liu et al., 2024a) incorporates ImageNet-based pretraining into a Mamba-based UNet for enhanced medical image segmentation performance. P-Mamba (Ye & Chen, 2024) combines Perona-Malik diffusion with Mamba to improve echocardiographic left ventricular segmentation in pediatric cardiology. Additionally, Wang *et al*. (Wang et al., 2024b) introduced LMa-UNet, a Mamba-based network with a large-window design for improved global context modeling.

Despite these advances, accurately segmenting complex medical images remains a challenge due to the intricate background and ambiguous class boundaries. Moreover, traditional scanning mechanisms, which intermittently scan different semantic regions, limit the model's ability to consistently capture the full range of contextual information within the images.

## 3 METHOD

In this section, we first introduce the foundational concepts pertinent to State Space Models (SSMs). Next, we provide a comprehensive overview of our proposed UD-Mamba architecture, with an overall framework illustrated in Figure 3. Finally, we elucidate the key components of UD-Mamba, detailing the operational workflow of the Uncertainty-Driven Selective Scanning Block (UD-SSB) and the derived optimization strategies.

### 3.1 PRELIMINARIES

In Mamba blocks, the token mixer operates as a specialized selective state space model (SSM) (Gu & Dao, 2023), which is characterized by its efficient handling of long-range dependencies through a compact memory representation. The model defines four core input parameters $(\Delta, \mathbf{A}, \mathbf{B}, \mathbf{C})$, which are transformed into $(\overline{\mathbf{A}}, \overline{\mathbf{B}}, \mathbf{C})$ using the following state-space dynamics:

$$\overline{\mathbf{A}} = \exp(\Delta A)$$
$$\overline{\mathbf{B}} = (\Delta \mathbf{A})^{-1}(\exp(\Delta \mathbf{A}) - \mathbf{I}) \cdot \Delta \mathbf{B} \tag{1}$$

The Mamba block excels in efficiently modeling temporal sequences using a structured state-space representation. The sequence transformation in the SSM is expressed as:

$$h_t = \overline{\mathbf{A}}h_{t-1} + \overline{\mathbf{B}}x_t$$
$$y_t = \mathbf{C}h_t \tag{2}$$

Here, $t$ refers to the temporal index, $x_t$ is the input sequence at time $t$, $h_t$ is the hidden state capturing the temporal context, and $y_t$ represents the output. The hidden state $h_t$ serves as a compact, memory-efficient repository that retains essential historical information, allowing the model to propagate context across time steps without increasing computational burden.

## 3.2 UD-MAMBA

The UD-Mamba architecture leverages a streamlined yet robust UNet framework (Ronneberger et al., 2015) with basic layers of Uncertainty-Driven (UD) Blocks. As illustrated in Figure 3, the design comprises three key components: a patch embedding layer that transforms the input image into a sequence of patches for subsequent processing, an encoder-decoder structure composed of UD blocks that captures and integrates both local and global features across varying scales, and a segmentation head that produces the final pixel-wise segmentation output based on the decoded features. The

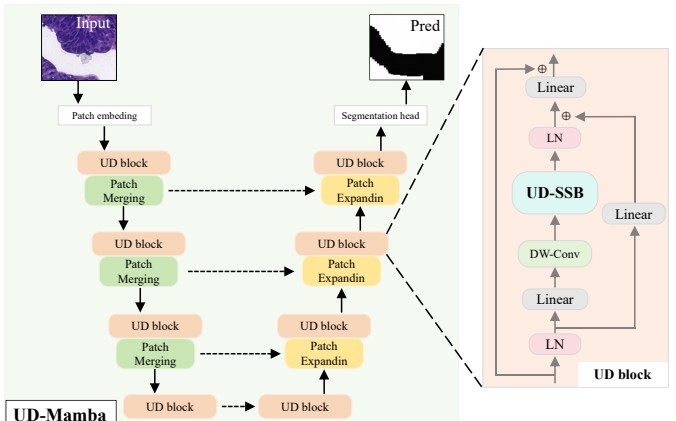

Figure 3: An illustration of UD-Mamba architecture.

encoder-decoder configuration is enhanced by skip connections, which facilitate the integration of multi-scale feature representations. Within the UD Block, Uncertainty-Driven Selective Scanning Block (UD-SSB) serve as the critical elements. This architectural choice enhances information propagation across levels, ultimately improving segmentation accuracy.

## 3.3 UNCERTAINTY-DRIVEN SELECTIVE SCANNING BLOCK

To address the limitations of traditional state-space models (SSMs) such as Mamba, which struggle with effectively modeling local features due to intermittent scanning of target regions, we propose a pixel-level uncertainty-driven selective scanning approach. This method is distinct from conventional pixel-order scanning mechanisms, as it leverages uncertainty at the pixel level to inform scanning sequences. As illustrated in Figure 4 I, our Uncertainty-Driven Selective Scanning Block (UD-SSB) introduces five key components: channel uncertainty computation, uncertainty-based sorting, scan expansion operations, the S6 block (Gu & Dao, 2023) processing, and the recovery operation.

Given an input feature tensor $\mathbf{X} \in \mathbb{R}^{B \times C \times H \times W}$, where $B$, $C$, $H$, and $W$ denote the batch size, number of channels, height and width respectively. We propose the following methodology:

**Channel uncertainty computation**: To compute an uncertainty map $\mathbf{U} \in \mathbb{R}^{B \times H \times W}$ for each spatial position across all channels, we define:

$$\mathbf{U} = \text{Uncertainty}(\mathbf{X}) \tag{3}$$

In this context, we utilize the standard deviation as our uncertainty metric, a choice validated by the results presented in Section 4.4.2. Specifically, for the input feature map $\mathbf{X} \in \mathbb{R}^{B \times C \times H \times W}$, we calculate the standard deviation across all channels $C$ for each spatial position $(h, w)$:

$$\mathbf{U}_{b,h,w} = \sqrt{\frac{1}{C} \sum_{c=1}^{C} (\mathbf{X}_{b,c,h,w} - \mu_{b,h,w})^2} \tag{4}$$

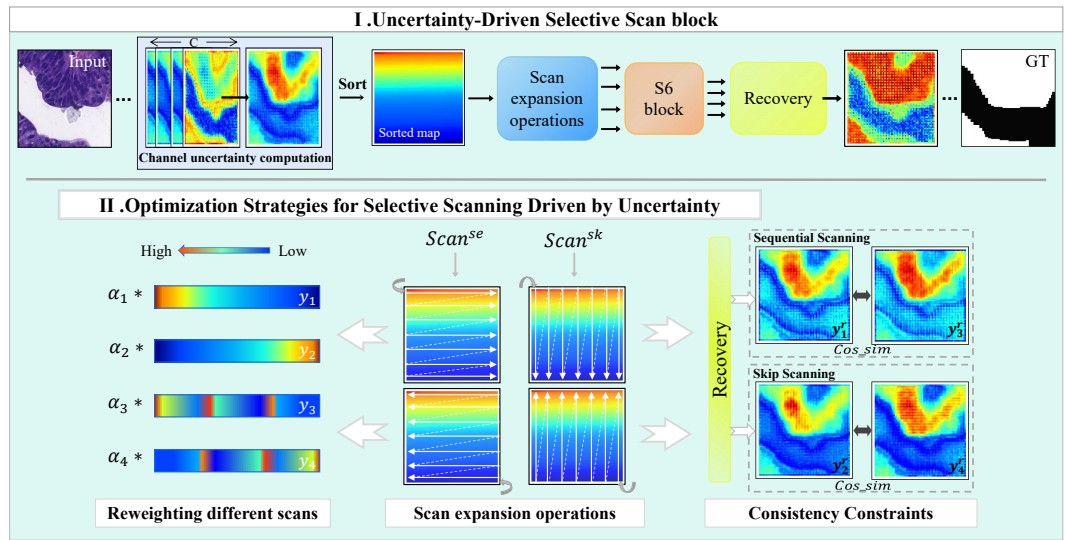

Figure 4: UD-SSB module architecture and optimization strategies. I. Depicts the main workflow of the UD-SSB module architecture. II. Illustrates our two optimization strategies for UD-SSB: reweighting different scans and cosine consistency constraint.

where $\mu_{b,h,w}$ represents the mean value at that spatial position across all channels. This calculation captures the pixel-level standard deviation across channels, where higher uncertainty typically corresponds to key regions, such as object boundaries or foreground regions, while lower uncertainty indicates background consistency. By focusing on pixel-level uncertainty, we can more precisely identify key regions for medical image segmentation, which is often critical when identifying pathological regions or organ boundaries.

**Uncertainty-based sorting**: The uncertainty map $\mathbf{U}$ is then sorted in descending order, resulting in $\mathbf{U}_{idx}$, which ranks the spatial locations from high-uncertainty regions (foreground and boundaries) to low-uncertainty regions (background). This allows the model to prioritize regions with higher complexity or importance during subsequent operations:

$$\mathbf{U}_{idx} = \text{Sort}(\mathbf{U}) \tag{5}$$

**Feature map rearrangement**: Using the sorted indices $\mathbf{U}_{idx}$, we rearrange the original feature map $\mathbf{X}$ to create $\mathbf{X}'$, where regions of high uncertainty are treated intensively. This reorganization prepares the feature map for efficient scanning:

$$\mathbf{X}' = \text{Rearrange}(\mathbf{X}, \mathbf{U}_{idx}) \tag{6}$$

**Scan expansion operations**: We implement two distinct scanning operations on the rearranged feature map $\mathbf{X}'$: 1) Sequential scanning ($\text{Scan}^{se}$): This operation processes spatial locations in descending order of pixel uncertainty, meaning that regions with higher uncertainty, such as foreground objects and boundaries, are prioritized. This approach ensures that key high-uncertainty regions are modeled intensively, allowing the model to focus on areas that are critical for accurate segmentation. 2) Skip scanning ($\text{Scan}^{sk}$): This operation selects spatial locations at regular intervals across the uncertainty spectrum, facilitating the interaction between background and foreground regions. By timely integrating background information, skip scanning helps maintain the overall background structure of the image while refining the details of the foreground, leading to more balanced segmentation results. The combination of sequential and skip scanning enables our model to effectively capture both local and global features.

**S6 block processing**: The scanned features are then processed by the S6 block (Gu & Dao, 2023):

$$S^{\text{out}} = \text{S6}(\text{Scan}^{se}(\mathbf{X}'), \text{Scan}^{sk}(\mathbf{X}')) \tag{7}$$

**Recovery operation**: Finally, the rearranged and processed features are restored to their original spatial configuration, ensuring that the spatial structure of the output remains consistent with the

input. This ensures that the model preserves positional information critical for accurate medical image segmentation:

$$\mathbf{X}_{\text{recovered}} = \text{Recover}(S^{\text{out}}) \tag{8}$$

### 3.4 UNCERTAINTY-DRIVEN SELECTIVE SCANNING OPTIMIZATION STRATEGY

As depicted in Figure 4 II, the UD-SSB applies four distinct scanning sequences: sequential and skip scans from high-to-low uncertainty levels ($\mathbf{y_1}$ and $\mathbf{y_2}$), and sequential and skip scans from low-to-high uncertainty levels ($\mathbf{y_3}$ and $\mathbf{y_4}$). Generally, regions with high uncertainty are more likely to correspond to target areas and critical boundaries, while low-uncertainty regions are usually associated with the background. In the Mamba framework, which operates as an autoregressive model, each output depends on the hidden state derived from previous inputs. Scanning from low-uncertainty to high-uncertainty regions allows the model to first process simpler background information, accumulating hidden state reserves before addressing more complex areas. As shown in Figure 2, this approach outperforms the reverse scanning order. Therefore, to capitalize on this property, we propose two optimization strategies to exploit the benefits of scanning from high-to-low uncertainty while mitigating the inherent limitations of scanning from low-to-high uncertainty.

#### 3.4.1 REWEIGHTING OF DIFFERENT SCANNING SEQUENCES

To optimize the contribution of each scanning sequence, we introduce four learnable parameters ($\alpha_1$, $\alpha_2$, $\alpha_3$, $\alpha_4$), each corresponding to one of the four scanning directions. These parameters serve to enhance the advantages of scanning from high-to-low uncertainty while modulating the contribution of each individual scanning sequence. The reweighting mechanism is mathematically defined as:

$$\mathbf{y_i'} = \mathbf{y_i} \cdot \alpha_i \quad \text{for} \quad i = 1, 2, 3, 4 \tag{9}$$

This approach ensures that each scanning method contributes in proportion to its effectiveness in capturing critical image regions, with greater emphasis placed on scans that progress from more certain to less certain areas.

#### 3.4.2 CONSISTENCY CONSTRAINTS BETWEEN BIDIRECTIONAL SCANS

To address the limitations associated with low-to-high uncertainty scans during the decoding phase and improve overall segmentation performance, we introduce a cosine consistency constraint at the end of the decoder. This constraint is applied between sequential and skip scans performed in both directions (from high-to-low and low-to-high uncertainty). By aligning the results from low-to-high uncertainty scans with those from high-to-low uncertainty scans, we ensure consistency in feature representation across different scanning directions. To maintain positional consistency, all outputs $\mathbf{y_i^r}$ are derived after the recovery operation is applied to $\mathbf{y_i}$. The cosine consistency loss is defined as:

$$L_{\cos} = 1 - \frac{\cos\_\text{sim}(\mathbf{y_1^r}, \mathbf{y_3^r}) + \cos\_\text{sim}(\mathbf{y_2^r}, \mathbf{y_4^r})}{2}, \tag{10}$$

where cos_sim represents the average cosine similarity between the forward and backward sequential and skip scans. By maximizing this similarity, we aim to minimize discrepancies between the two scanning directions, thereby reinforcing the consistency of the final segmentation outputs.

Finally, the overall loss function combines the supervised loss with the cosine consistency loss:

$$L = L_{\text{sup}} + \lambda \cdot L_{\cos}, \tag{11}$$

where $L_{\text{sup}}$ represents the combined cross-entropy and Dice loss (CeDice loss), $L_{\cos}$ is the cosine similarity loss, and $\lambda$ is a hyperparameter that balances these two components.

## 4 EXPERIMENTS

### 4.1 DATASET

To verify the effectiveness of UD-Mamba, we comprehensively evaluate its performance on three medical image datasets: DigestPath, ISIC 2018 and ACDC.

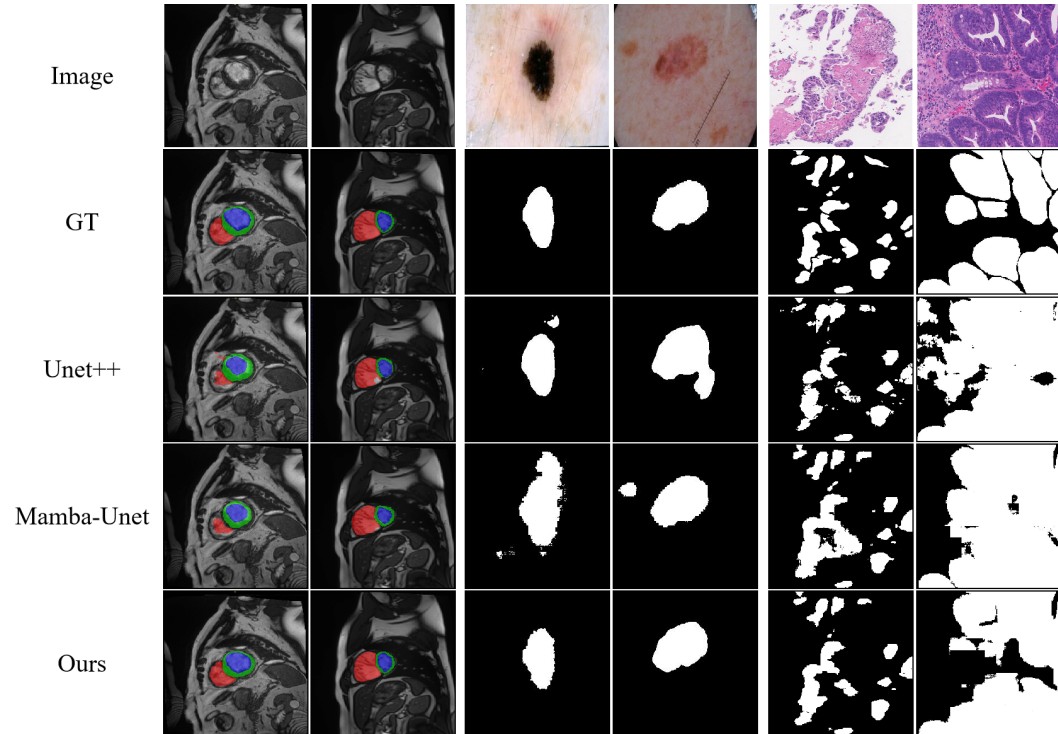

Figure 5: Visual comparisons of segmentation results from UD-Mamba and various other methods are conducted across three different datasets.

The DigestPath dataset (Da et al., 2022) comprises whole slide images (WSIs) for binary segmentation of tumor lesions in colonoscopy. We randomly divided 130 malignant WSIs into three subsets: 100 for training, 10 for validation, and 20 for testing. For model training, WSIs were further partitioned into 256 × 256 pixel patches, yielding a training set of 29,412 patches. Our model evaluation was conducted at the WSI level.

The ISIC 2018 dataset (Codella et al., 2019), part of the 2018 International Skin Imaging Collaboration challenge, is a public dataset for skin lesion segmentation containing 2,694 dermoscopy images with corresponding label data. We split the dataset into training, validation, and test sets using a 7:2:1 ratio. Based on these two datasets, we conducted a detailed evaluation using performance metrics including mean Intersection over Union (mIoU), Dice Similarity Coefficient (DSC), Accuracy (Acc), Sensitivity (Sen), and Specificity (Spe).

The ACDC dataset (Bernard et al., 2018) consists of cardiac cine MRI scans from 100 patients, used for the segmentation of three cardiac substructures: the Left Ventricle (LV), Right Ventricle (RV), and Myocardium (MYO). We split the dataset into 70% for training, 10% for validation, and 20% for testing. All slices were resized to a uniform resolution of 256 × 256 pixels before training. Performance was evaluated using the Dice Similarity Coefficient (DSC), mean Intersection over Union (mIoU), and 95% Hausdorff Distance ($HD_{95}$). Given the fixed anatomical structures in ACDC, the inclusion of $HD_{95}$ provides a more robust assessment of boundary accuracy.

### 4.2 IMPLEMENTATION DETAILS

All experiments were conducted using the PyTorch framework on an Ubuntu desktop equipped with an NVIDIA RTX A6000 GPU. Training was performed using Stochastic Gradient Descent (SGD) with a multi-step learning rate strategy, initially set to 0.01. The total number of training epochs was fixed at 300. For UD-Mamba, each layer of both the encoder and decoder corresponds to two UD blocks. We utilize weights pre-trained on ImageNet-1K (Deng et al., 2009) to initialize the encoder.

## 4.3 COMPARISON WITH EXISTING METHODS

To validate the effectiveness of our proposed UD-Mamba model, we compared it with state-of-the-art medical image segmentation methods across three datasets: ISIC, DigestPath, and ACDC. Specifically, the models evaluated included CNN-based approaches (such as UNet (Ronneberger et al., 2015), UNet++ (Zhou et al., 2019) and Att-UNet (Oktay et al., 2018)), Transformer-based models (like TransUNet (Chen et al., 2024) and SwinUNet (Cao et al., 2022)), as well as Mamba-based models (Mamba-UNet (Wang et al., 2024c)). The visualization results are shown in Figure 5.

Table 1: Performance comparison of different networks on ISIC 2018 and Tissue datasets.

| Dataset | ISIC 2018 | | | | | DigestPath | | | | |
|---|---|---|---|---|---|---|---|---|---|---|
| Network | DSC(%)↑ | IoU(%)↑ | ACC(%)↑ | Spe(%)↑ | Sen(%)↑ | DSC(%)↑ | IoU(%)↑ | ACC(%)↑ | Spe(%)↑ | Sen(%)↑ |
| UNet | 86.51 | 77.81 | 92.91 | 94.90 | 88.69 | 77.96 | 64.91 | 94.30 | 96.09 | 80.74 |
| UNet++ | 87.36 | 79.20 | 93.10 | 95.59 | 88.71 | 78.37 | 65.43 | 94.52 | 96.13 | 80.41 |
| TransUNet | 88.12 | 80.32 | 93.91 | 94.04 | 89.40 | 79.30 | 66.74 | 94.64 | 96.27 | 81.18 |
| SwinUNet | 87.20 | 79.27 | 93.49 | 96.22 | 87.30 | 79.15 | 66.54 | 94.75 | 96.84 | 79.98 |
| Att-UNet | 87.47 | 79.31 | 93.12 | 95.77 | 88.83 | 78.28 | 65.24 | 94.38 | 95.78 | 81.57 |
| Mamba-UNet | 87.86 | 80.36 | 93.79 | 96.36 | 89.61 | 79.92 | 67.41 | 94.65 | 96.06 | 82.47 |
| Ours | 89.15 | 81.94 | 94.60 | 96.26 | 89.55 | 80.89 | 68.64 | 94.98 | 96.44 | 83.34 |

For the ISIC 2018 and DigestPath datasets, as shown in Table 1, we employed five evaluation metrics to assess the model's segmentation performance. Firstly, the UD-Mamba method significantly outperforms CNN-based approaches. Specifically, UD-Mamba achieved improvements of 1.68% and 2.52% in DSC over the best CNN methods on the ISIC 2018 and DigestPath datasets, respectively. Moreover, the mIoU scores increased by 2.63% and 3.21%. Compared to Transformer-based models such as TransUNet (Chen et al., 2024), our method demonstrated a notable advantage in mIoU, with increases of 1.62% for ISIC 2018 and 1.90% for DigestPath. Additionally, when compared to the representative Mamba-based model Mamba-UNet (Wang et al., 2024c), UD-Mamba improved the mIoU by 1.58% and 1.23% on the two datasets, respectively.

Table 2: Comparison of different networks on ACDC dataset.

| Network | DSC(%)↑ | RV | MYO | LV | mIoU(%)↑ | $HD_{95}(mm)$↓ |
|---|---|---|---|---|---|---|
| UNet (Ronneberger et al., 2015) | 90.07 | 89.11 | 87.22 | 93.89 | 82.42 | 2.74 |
| UNet++ (Zhou et al., 2019) | 90.23 | 89.08 | 87.65 | 93.96 | 82.64 | 1.90 |
| TransUNet (Chen et al., 2024) | 90.70 | 91.71 | 87.74 | 92.68 | 83.50 | 2.76 |
| SwinUNet (Cao et al., 2022) | 89.45 | 90.52 | 86.23 | 91.60 | 81.55 | 3.56 |
| Att-UNet (Oktay et al., 2018) | 89.17 | 88.45 | 86.14 | 92.94 | 81.04 | 3.17 |
| Mamba-UNet (Wang et al., 2024c) | 91.08 | 90.80 | 88.09 | 94.35 | 84.03 | 1.40 |
| Ours | 91.99 | 90.85 | 90.69 | 94.45 | 85.48 | 1.31 |

For the ACDC dataset, Table 2 presents a comparison of results with other methods. Compared to the best-performing Mamba-UNet (Wang et al., 2024c), our approach demonstrated significant improvements, with increases of 0.91% and 1.45% in DSC and mIoU, respectively, while reducing the $HD_{95}$ metric to 1.31 mm.

Table 3: Comparison of the effects of uncertainty scanning and its optimization strategies.

| UD-SSB | Reweight | $L_{cos}$ | DSC ↑ | mIoU ↑ | ACC ↑ |
|---|---|---|---|---|---|
| | | | 79.92 | 67.41 | 94.65 |
| ✓ | | | 80.32 | 68.14 | 94.93 |
| ✓ | | ✓ | 80.41 | 68.18 | 94.94 |
| ✓ | ✓ | | 80.72 | 68.55 | 94.97 |
| ✓ | ✓ | ✓ | **80.89** | **68.64** | **94.98** |

Table 4: Comparison of different methods for measuring the uncertainty of channels.

| Method | DSC ↑ | mIoU ↑ | ACC ↑ |
|---|---|---|---|
| Mad | 80.75 | 68.58 | 94.97 |
| Range | 79.89 | 68.07 | 94.88 |
| Entroph | 80.28 | 67.96 | 94.89 |
| Variance | 80.02 | 67.26 | 94.75 |
| STD | **80.89** | **68.64** | **94.98** |

## 4.4 ABLATION STUDIES

In the ablation study section, we conduct experimental verification on the DigestPath dataset (Da et al., 2022).

### 4.4.1 ABLATION OF UNCERTAINTY SCANNING AND ITS OPTIMIZATION STRATEGIES

We conducted ablation experiments on the pixel-level channel uncertainty-driven scanning operation and its optimization strategies. As shown in Table 3, without our components (first row), the method degenerates into a standard position-based scanning approach. We observed that the uncertainty-driven scanning method yielded superior results compared to the original scanning method (DSC: 80.32% vs. 79.92%). This confirms that the uncertainty-driven scanning approach effectively separates uncertain regions representing foreground and boundaries from background-related areas, thereby enhancing local modeling capabilities. As illustrated in Figure 6, our method demonstrates excellent modeling capability for target regions compared to the traditional position-based scanning method used in Mamba-UNet (Wang et al., 2024c). Further-

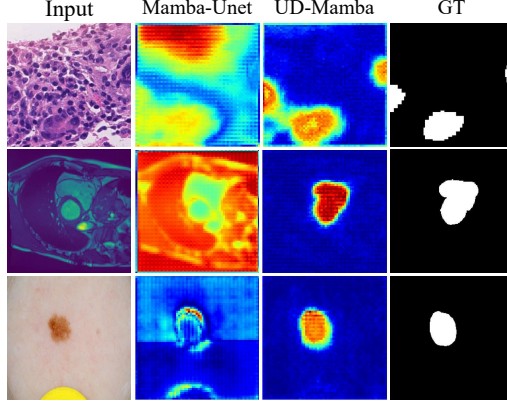

Figure 6: Visual comparisons of uncertainty maps.

more, the re-weighting and consistency constraint strategies further enhance the model's representational capacity. These strategies amplify the advantages of scanning from high to low uncertainty while mitigating the limitations of scanning from low to high uncertainty, resulting in an improved DSC of 80.89%.

### 4.4.2 ABLATION OF DIFFERENT UNCERTAINTY CALCULATION METHODS

To evaluate various criteria for measuring channel uncertainty, we conducted ablation experiments. These criteria include Mean Absolute Deviation (MAD), Standard Deviation (STD), Variance, Entropy and the Range between the two highest values. As illustrated in Table 4, the use of STD provides a stable measure of data dispersion. This stability enables the model to more reliably identify true regions of uncertainty, rather than being misled by noise or outliers. Consequently, the method that employs STD to calculate uncertainty achieved the best results, attaining the highest Dice Similarity Coefficient (DSC) of 80.89%.

### 4.4.3 ABLATION OF UNCERTAINTY CALCULATION REGION

To evaluate the effectiveness of pixel-level uncertainty-driven scanning in scenarios lacking explicit spatial features, we conducted comparative experiments focusing on the size of the regions used for uncertainty calculations. Instead of relying solely on the uncertainty of individual pixels, we extended the calculation

Table 5: Comparison of different methods for calculating uncertainty.

| Size | Static | | | | Dynamic | |
|------|--------|--------|--------|--------|----------------|----------------|
| | 1 | 2 | 4 | 8 | $a_v/a_v^{min}$ | $a_v^{max}/a_v$ |
| DSC ↑ | 80.89 | 80.44 | 80.19 | 79.82 | 79.85 | 79.93 |

to larger regions to retain some degree of spatial information. These regions are defined as uncertainty blocks with dimensions $a \times a$.

Our experimental design explores both fixed and dynamically adjusted values for $a$. For fixed-size regions, we varied $a$ from 1 up to $a_v^{min}$. In the case of dynamically adjusted regions, two strategies were employed: (1) proportional scaling, where $a = a_v/a_v^{min}$, allowing $a$ to increase proportionally with the feature vector size $a_v$; and (2) inverse proportional scaling, where $a = a_v^{max}/a_v$, causing $a$ to decrease as the feature vector size $a_v$ increases. Here, $a_v$ refers to the feature vector size at each stage before entering the UD-SSB, $a_v^{max}$ represents the feature vector size upon the first entry into the UD-SSB, and $a_v^{min}$ denotes the feature vector size at the bottleneck layer. In UD-Mamba, $a_v^{max}$ and $a_v^{min}$ are set to 64 and 8, respectively. After calculating the average uncertainty value for each region, these values are used to rank the regions for subsequent scanning. As demonstrated in Table 5, pixel-level uncertainty-driven scanning consistently outperforms both dynamic and static region-based methods. This result highlights the advantages of pixel-level granularity in determining uncertainty for fine-grained tasks like medical image segmentation. Compared to broader region-based uncertainty approaches, pixel-level uncertainty focuses on capturing local variations, providing a more precise method for identifying critical segmentation targets.

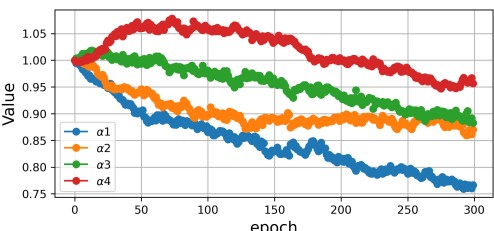 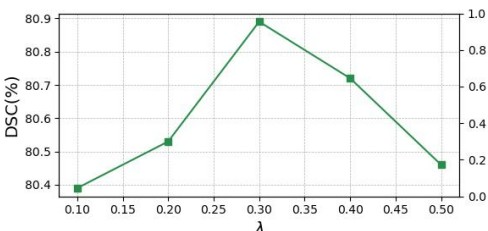

Figure 7: Analysis of recorded values for four learnable reweighting parameters.

Figure 8: Sensitivity analysis of the hyperparameter $\lambda$.

### 4.4.4 CHANGES IN RE-WEIGHTING VALUES FOR FOUR DIFFERENT SCANNING METHODS

Figure 7 illustrates the evolution of the four learnable parameters $\alpha_1$, $\alpha_2$, $\alpha_3$, and $\alpha_4$, which reweight the four different scanning sequences, throughout the training process. All four parameters exhibit a downward trend, with $\alpha_3$ and $\alpha_4$ showing a less pronounced decrease compared to $\alpha_1$ and $\alpha_2$. This pattern suggests that during training, the scanning processes from high to low uncertainty levels, corresponding to $\alpha_3$ and $\alpha_4$, contribute more significantly than the scanning processes from low to high uncertainty levels associated with $\alpha_1$ and $\alpha_2$. This observation indirectly corroborates the conclusion proposed in Figure 2.

### 4.4.5 ABLATION OF HYPERPARAMETER $\lambda$

For the hyperparameter $\lambda$, which controls the magnitude of the consistency constraint loss between bidirectional scans, we conducted ablation experiments to determine its optimal range. As shown in Figure 8, the best results were obtained when $\lambda$ was set to 0.3.

## 5 CONCLUSION

In this paper, we introduce UD-Mamba, a novel architecture designed to address Mamba's limitations in local feature modeling. By integrating a pixel-level channel uncertainty-driven mechanism, UD-Mamba effectively prioritizes pixels based on channel uncertainty, enabling comprehensive and efficient feature extraction. Furthermore, as scanning from low-uncertainty to high-uncertainty vectors typically yields greater benefits than the reverse process, we introduce four learnable parameters to explore the impact of various scanning sequences on the autoregressive Mamba framework. Concurrently, we enhance the efficacy of transitions from high-uncertainty to low-uncertainty regions by constraining the cosine similarity loss between forward and backward scanning results. Experimental results on three medical imaging datasets demonstrate UD-Mamba's superior performance in medical image segmentation tasks compared to traditional models.

Future work will focus on developing more precise and effective uncertainty estimation methods, as model performance depends heavily on accurate channel uncertainty estimation. The use of standard deviation to evaluate uncertainty may be inadequate for capturing more complex patterns across diverse medical imaging tasks. Additionally, we aim to expand the application of UD-Mamba to a wider range of medical image segmentation challenges.

**Ethical Considerations**: All authors of this paper have reviewed and are committed to upholding the ethical guidelines outlined in the ICLR Code of Ethics.

**Reproducibility Statement**: We provide a detailed description of the model architecture (UD-Mamba in Section 3.2), loss functions (Section 3.4.2), and training procedures (Section 4.2). All related code will be open-sourced to ensure full reproducibility.

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
