# OpenReview forum: "UD-Mamba: A pixel-level uncertainty-driven mamba model for medical image segmentation"
_ICLR.cc/2025/Conference — ICLR 2025 Conference Withdrawn Submission_

### Official Review · Reviewer_LtcA · 2024-10-31

**Soundness:** 3
**Presentation:** 3
**Contribution:** 2
**Rating:** 5
**Confidence:** 4

**Summary:**

The paper proposed a modification to MAMBA for medical image segmentation. They propose an uncertainty scanning approach to collect the sequential spatial data. They test their method on 3 datasets, in an effort to showcase how their method outperforms other SoTA methods.

**Strengths:**

It is interesting to examine alternative scanning approaches, and to test on multiple dataset.

**Weaknesses:**

... Though the chosen scanning strategy is not strongly and theoretically motivated. The results show improvement but these are too small and may turn out to be statistically insignificant.

**Questions:**

High-level review:

It is interesting to examine alternative scanning approaches, and to test on multiple dataset. Though the chosen scanning strategy is not strongly and theoretically motivated. The results show improvement but these are too small and may turn out to be
statistically insignificant.


Detailed questions and notes:


Different channels capture different features. I don't understand why varying features would be called uncertainty. The author even refers to this so-called uncertainty as complexity or importance. I would call it what it is, e.g., feature diversity.

Why not learn the optimal scanning trajectory, similar to the approach in "Predicting Cancer with a Recurrent Visual Attention Model for Histopathology Images" https://doi.org/10.1007/978-3-030-00934-2_1 (though that was for classification)?

L261: There is an underlying assumption in this statement that the background and foreground are at opposite ends of the uncertainty spectrum. Is this truly the case? In some medical images, the foreground object may be homogeneous (e.g., almost constant intensity organ tissue) while the background is complex (all other organs and tissues in the body).

What is y_i^r on L306? Is that supposed to be  y_i' ?

What is the dimensionality of  y_i ?

L310-312: How does maximizing (10) minimize discrepancy and reinforce consistency? Wouldn't minimizing this loss result in y_1 and y_3 being aligned make low-to-high-uncertainty scanning equal to high-to-low-uncertainty scanning? This would drive the channels to have features with constant uncertainty, thus low diversity. Why would this be a good thing?

How do you handle the case when all alphas are zeros? Shouldn't you have a constraint that some norm of the vector of alphas be set to unity, for example,  |[ alpha_1, alpha_3 ] | = 1  or | [ alpha_1, alpha_2, alpha_3, alpha_4 ] | = 1 ?

Why do the first two columns of Fig. 5 have a different visualization than the other columns? I suggest showing both GT and segmentation on the same figure (e.g., semi-transparent or using contours), as it is hard to compare segmentation with GT (where they agree and where they don't) when they are in different figures.

Table 1: Highlight (in shading or bold) the highest/best value in each column. The shading at the bottom gives the impression that "Ours" is the best for all columns when it is not. For example, Mamba-UNet has higher Sep(%) and Sen(%) for ISIC2018, and SwinUNET has higher Sep(%) in DigestPath. The same applies to Table 2, where TranUNET has higher DSC for RV.

What is the uncertainty colorbar for Fig. 1 and 6? Is that a jet colorbar, where the highest is red and the lowest is blue?

Table 3: The improvements (especially L_cos) are minimal. Are they statistically significant? This observation of minor improvement, which may not be statistically significant, applies to the results in other tables as well.

It is important to observe how the trend continues in Figure 7.

Was lambda optimized on a validation set and then fixed for the test set?

---

### Official Review · Reviewer_weXe · 2024-11-01

**Soundness:** 2
**Presentation:** 2
**Contribution:** 2
**Rating:** 3
**Confidence:** 4

**Summary:**

The manuscript introduces a novel approach to medical image segmentation by leveraging a pixel-level uncertainty-driven mechanism within the Mamba framework. The proposed UD-Mamba defines the scanning process to prioritize regions of high uncertainty, leading to improved segmentation precision.

**Strengths:**

The introduction of pixel-level uncertainty-driven scanning is important for medical image segmentation for handling complex and ambiguous boundaries. Overall the method is well-constructed and illustrated in the article.

**Weaknesses:**

1. The experiments are only constructed on 2D images. While most of medical images like CT and MRI are 3D images, which is significantly different from natural images. How to extend the method to 3D datasets? Besides, ACDC is a 3D MRI dataset, what is the experimental design? Is the segmentation performed slice by slice?
2. Lack of comparisons with several state-of-the-art methods like nnUNet, STUNet, U-Mamba.
3. There is a need for a deeper theoretical justification for the choice of uncertainty metrics and the impact of the scanning strategies on the overall performance of the model.

**Questions:**

Although the proposed method sounds acceptable. The experiments are only conducted on limited datasets and lack of comparison with state-of-the-art methods. Besides, the efficiency like parameters/runing time of proposed method is not shown.

---

### Official Review · Reviewer_iHXH · 2024-11-02

**Soundness:** 2
**Presentation:** 2
**Contribution:** 2
**Rating:** 5
**Confidence:** 4

**Summary:**

This paper introduces a novel medical image segmentation framework called “Uncertainty-Driven Mamba.” The authors propose an Uncertainty-Driven Selective Scanning Block and an Uncertainty-Driven Selective Scanning Optimization Strategy to enhance Mamba-based segmentation methods. Experimental results imply that this framework achieves good performance.

**Strengths:**

1. The writing is well and easy to follow.
2. The idea of Uncertainty-Driven Mamba is interesting.
3. The code is available (on supplementary).

**Weaknesses:**

1. The compared methods used in this paper are outdated. With the exception of Mamba-UNet on arXiv, all other methods were **originally** proposed in 2021 or earlier. At least the authors should include comparisons with some more recent, well-cited medical segmentation methods, such as SS-Former [*1], H2Former [*2], PVT-CASCADE [*3], Medsegdiff-v2 [*4].
2. The results appear to lack reproducibility. For each dataset, it seems that the authors use their own **random** dataset splits instead of following the widely adopted standard benchmarks. For instance, in skin lesion segmentation, most studies (such as [*5, *6]) employ a 7:3 split for training and testing data, whereas this paper uses a customised 7:1:2 split. Given that these standard splits are publicly available, there seems to be no need to deviate from widely used benchmarks.
3. As the title refers to “medical image segmentation,” the readers would like to see results on datasets that have also been widely evaluated in other studies [*1, *2, *3]. Please consider using the polyp segmentation benchmark proposed in [*7].
4. Efficiency is important in medical image segmentation. Please provide details on the parameters, FLOPs, and FPS of the different methods for a comprehensive comparison.
5. In the last column of Figure 5, the results of UD-Mamba is also not good (although a little better than other methods).

[*1] Stepwise Feature Fusion: Local Guides Global, MICCAI 2022

[*2] H2Former: An Efficient Hierarchical Hybrid Transformer for Medical Image Segmentation, IEEE TMI 2023

[*3] Medical Image Segmentation via Cascaded Attention Decoding, WACV 2023

[*4] Medsegdiff-v2: Diffusion-based Medical image Segmentation with Transformer, AAAI 2024

[*5] MALUNet: A Multi-Attention and Light-weight UNet for Skin Lesion Segmentation, IEEE BIBM 2022

[*6] EGE-UNet: An Efficient Group Enhanced UNet for Skin Lesion Segmentation, MICCAI 2023

[*7] Pranet: Parallel Reverse Attention Network for Polyp Segmentation, MICCAI 2020

**Questions:**

See weakness.

---

### Note · Authors · 2024-11-15

I have read and agree with the venue's withdrawal policy on behalf of myself and my co-authors.